# Cucumber Mildew Resistance Locus O Interacts with Calmodulin and Regulates Plant Cell Death Associated with Plant Immunity

**DOI:** 10.3390/ijms20122995

**Published:** 2019-06-19

**Authors:** Guangchao Yu, Xiangyu Wang, Qiumin Chen, Na Cui, Yang Yu, Haiyan Fan

**Affiliations:** 1College of Horticulture, Shenyang Agricultural University, Shenyang 110866, China; yugc7674@163.com (G.Y.); wxykids@163.com (X.W.); QiuminChen2019@163.com (Q.C.); 2College of Bioscience and Biotechnology, Shenyang Agricultural University, Shenyang 110866, China; syaua@163.com (N.C.); yy7603@163.com (Y.Y.); 3Key Laboratory of Protected Horticulture of Ministry of Education, Shenyang Agricultural University, Shenyang 110866, China

**Keywords:** *Cucumis sativus*, CsMLO1 genes, *CsCaM3* genes, cell death, plant defense, expression analysis, *Corynespora cassiicola*

## Abstract

Pathogen-induced cell death is closely related to plant disease susceptibility and resistance. The cucumber (*Cucumis sativus* L.) mildew resistance locus O (CsMLO1) and calmodulin (CsCaM3) genes, as molecular components, are linked to nonhost resistance and hypersensitive cell death. In this study, we demonstrate that CsMLO1 interacts with CsCaM3 via yeast two-hybrid, firefly luciferase (LUC) complementation and bimolecular fluorescence complementation (BiFC) experiments. A subcellular localization analysis of green fluorescent protein (GFP) fusion reveals that *CsCaM3* is transferred from the cytoplasm to the plasma membrane in *Nicotiana benthamiana*, and *CsCaM3* green fluorescence is significantly attenuated via the coexpression of *CsMLO1* and *CsCaM3. CsMLO1* negatively regulates *CsCaM3* expression in transiently transformed cucumbers, and hypersensitive cell death is disrupted by *CsCaM3* and/or *CsMLO1* expression under *Corynespora cassiicola* infection. Additionally, *CsMLO1* silencing significantly enhances the expression of reactive oxygen species (ROS)-related genes (*CsPO1*, *CsRbohD*, and *CsRbohF*), defense marker genes (*CsPR1* and *CsPR3*) and callose deposition-related gene (*CsGSL*) in infected cucumbers. These results suggest that the interaction of CsMLO1 with CsCaM3 may act as a cell death regulator associated with plant immunity and disease.

## 1. Introduction

In natural environments, plants are exposed to attack from a variety of microbial pathogens. Therefore, plants have acquired complex innate immune systems to combat long-term biotic threats, thereby reducing the damage caused by pathogen invasion. The evolutionary process of the plant–pathogen interaction is divided into four stages. In the first stage, plant pattern-recognition receptors (PRRs) recognize conserved pathogen-associated molecular patterns (PAMPs) of microorganisms, and PAMP-triggered immunity (PTI) is usually strong enough to prevent the colonization of most microbial pathogens. In the second stage, evolved pathogens secrete virulence factors that inhibit PTI to cause effector-triggered susceptibility (ETS) in plants. In the third stage, plants express a specific *R* gene to directly or indirectly recognize pathogen-specific effector factors, and then effector-triggered immunity (ETI) accelerates and amplifies PTI to cause disease resistance in plants. In the fourth stage, under the pressure of natural selection, the pathogen is forced to produce new effector factors to inhibit ETI, although plants also produce new R genes to activate ETI to maintain their survival [1,2,3]. Upon recognition of certain effector proteins, host cells undergo rapid programmed cell death, namely, the hypersensitive response (HR), which in turn inhibits pathogen infection [4].

Most plants are resistant to complete species of microbial invaders, a phenomenon that is known as nonhost resistance (NHR) [5]. NHR provides broad-spectrum and strong resistance to nonadapted pathogens in plants [6,7,8]. Currently, the mildew resistance locus O (MLO) gene in nonhost defense plays an important role in the model system of the interaction of barley with *Blumeria graminis* f. sp. *hordei* (*Bgh*) [9,10]. Many reports have found that barley *mlo* mutant plants are almost completely resistant to powdery mildew pathogens because the host cell wall thickens to prevent the penetration of powdery mildew [6,10,11]. Additionally, the three penetration proteins PEN1, PEN2, and PEN3, which are closely related to the MLO gene defense, are central components of cell wall-based defense against nonadapted powdery mildew [12]. In our laboratory, transcriptome and isobaric tags for relative absolute quantification (iTRAQ) analyses identified two genes, *CsMLO1* and *CsMLO2*, in cucumber that are involved in the response to *Corynespora cassiicola*. Proteins/genes that are mainly involved in the defense response and oxidative stress and calcium signaling pathways during *C. cassiicola* infection have been identified in cucumber [13,14]. In addition, our research finds that the functions of *CsMLO1* and *CsMLO2* in *C. cassiicola* infection act as negative modulators to enhance the expression of reactive oxygen species (ROS)-related genes and defense-related genes for improved cucumber disease resistance [15]. However, few studies have focused on the resistance mechanism of MLO in cucumbers against *C. cassiicola* infection.

Hypersensitive cell death leads to the complex activation of defense signaling pathways, as the hallmarks of plant resistance, including the activation of ROS and defense genes, changes in the intercellular calcium levels, and modifications of cell walls and callose deposition [16,17]. The results of genetic epistasis analyses suggest that pathogen-responsive callose deposition has been detected in *Atmlo2* mutant plants [18]. *Arabidopsis MLO2* is a negative regulator of plant ROS responses to biotic stress [19]. In cucumber transgenic cotyledons, the transcription level of HR-associated genes is negatively regulated by *CsMLO1* expression [15]. In addition, reports have shown that a calmodulin-binding domain in the cytoplasmic carboxy-terminal region of the MLO protein interacts with Ca^2+^-dependent calmodulin (CaM) [20,21]. The function of CaM has been extensively studied as an HR-related gene. For instance, after exposure to *Xanthomonas campestris* pv. *vesicatoria, Agrobacterium* spp.-mediated *CaCaM1* overexpression activated ROS, nitric oxide (NO) generation, and HR-like cell death in pepper leaves, which exhibited enhanced resistance [22,23,24]. Furthermore, an analysis of a calcium inhibitor demonstrated that the inhibitor obstructed *CaCaM1*-triggered cell death, thus showing that *CaCaM1*-induced cell death depended on cytosolic Ca^2+^ [25,26]. Therefore, we hypothesize that there is a certain regulatory relationship between *CsMLO* and *CsCaM* that affects the resistance of cucumber to *C. cassiicola* infection.

In our laboratory, *CsMLO1* (XM_004148737) and *CsMLO2* (XM_004142345) of cucumber have been shown to be involved in the response to *C. cassiicola.* Phenotypic analyses of transgenic cucumber cotyledons inoculated with *C. cassiicola* revealed that transient overexpression of either *CsMLO1* or *CsMLO2* in cucumber cotyledons reduced resistance to *C. cassiicola*, whereas silencing of either *CsMLO1* or *CsMLO2* enhanced resistance to *C. cassiicola*. [13,14]. In addition, significant changes of calmodulin involved in calcium signaling were detected after treatment with *C. cassiicola* by iTRAQ analysis. However, the interaction between MLO and CaM in cucumber has not been precisely demonstrated, and the mechanism underlying the expression regulation between MLO and CaM is poorly understood. In this study, the analysis demonstrated that CsMLO1 stably interacted with CsCaM3 via yeast two-hybrid, firefly luciferase (LUC) complementation and bimolecular fluorescence complementation (BiFC) experiments. The coexpression of *CsMLO1* and *CsCaM3* in tobacco cells revealed that cytosolic *CsCaM3* was transferred to the plasma membrane, where it colocalized with *CsMLO1*, which resulted in a decrease in the fluorescence of *CsCaM3*. Additionally, *CsMLO1,* as a negative modulator, can not only enhance the defense response to cucumber-*C. cassiicola* interactions, but also negatively regulate the expression of *CsCaM3.* Furthermore, *CsMLO1* and *CsCaM3* negatively regulated hypersensitive cell death, including ROS burst, cell death, and defense-related gene expression, under *C. cassiicola* infection. These observational mechanisms suggested that CsMLO1 interacted with CsCaM3 and negatively regulated *CsCaM3* expression, thereby inhibiting plant immune responses.

## 2. Results

### 2.1. CsMLO-Mediated Regulation of CsCaM Expression Patterns in Cucumber Cotyledons

*CsMLO1* and *CsMLO2* were isolated previously and functionally characterized in *C. cassiicola*-infected cucumber leaves [15]. The sequences of calmodulin-7 (A0A0A0KWT3) were aligned with the cucumber genome database using the service provided by http://cucurbitgenomics.org/BLAST. Finally, we screened three of the most similar cucumber calmodulin genes: *CsCaM1* (XM_011655459), *CsCaM2* (XM_004144051), and CaM3 (XM_004142130). To further confirm these results, we determined the expression of cucumber calmodulin genes (*CaM1*, *CaM2*, and *CaM3*) in response to exposure to *C. cassiicola* for different time periods (Appendix A). Reverse-transcription quantitative PCR (RT-qPCR) assays showed that the expression patterns of *CaM1*, *CaM2*, and *CaM3* were upregulated in the defense response of *C. cassiicola*, although *CsCaM3* expression was rapidly and strongly induced in the early stages of *C. cassiicola* invasion. These results suggested that *CsCaM3* might be a positive regulatory modulator that participates in the immunity of cucumber to *C. cassiicola* in the early stages of infection.

The results showed that the transcriptional levels of *CsMLO1*, *CsMLO2*, and *CsCaMs* changed after *C. cassiicola* infection, which in turn affected the defense response of cucumber to pathogens. These genes functioned in the defense response to *C. cassiicola*; thus, whether the *CsMLO1* and *CsMLO2* genes affected the transcription levels of *CsCaM1*, *CsCaM2*, and *CsCaM3* was further investigated. In a previous study, an experimental method for the transient agroinfiltration of cucumber cotyledons was established [14,27,28]. Here, a fragment 387 bp in length between the *CsMLO1/CsMLO2* and the green fluorescent protein (GFP) fusion genes was detected via PCR analysis in cucumber cotyledons transiently overexpressing *CsMLO1/CsMLO2* (Figure 1A). The RT-qPCR assay showed that the expression of the *CsMLO1/CsMLO2* genes was higher in the transgenic cucumbers than in the empty-vector control cucumbers (Figure 1B). Furthermore, the GFP imaging assay confirmed the success of the transient overexpression of *CsMLO1/CsMLO2* in cucumber protoplasts (Figure 1C). *CsMLO1*-GFP fluorescence and *CsMLO2*-GFP fluorescence were located in the plasma membrane, while the GFP control was predominantly located in the cytosol and membrane. In addition, the localization of CsMLO1 and CsMLO2 appeared solely in the plasma membrane in *Nicotiana benthamiana* leaf cells (Appendix A). The above results further demonstrated that *CsMLO1* and *CsMLO2* overexpression was successful in transgenic cucumber cotyledons, and they also showed for the first time that the genes were primarily localized in the plasma membrane of cucumber protoplasts.

Moreover, we used tobacco rattle virus (TRV)-based virus-induced gene silencing (VIGS) to silence *CsMLO1*/*CsMLO2* transcripts in cucumber cotyledons (Figure 2). After an *Agrobacterium* suspension (TRV::00, TRV::*CsMLO1*, and TRV::*CsMLO2*) was injected into cucumber cotyledons for 10 days, the chlorotic mosaic symptoms of TRV emerged in the cotyledons of transgenic cucumber plants, while no symptoms appeared in the control, suggesting that TRV successfully invaded the cucumber plants (Figure 2A). The efficient silencing of *CsMLO1* and *CsMLO2* was confirmed by RT-qPCR (Figure 2B). These results indicated that expression was almost abolished in the *CsMLO1*-silenced and *CsMLO2*-silenced cucumbers. To explore whether the expression of HR-related genes (*CsCaMs*) was regulated by *CsMLO1*/*CsMLO2*, we analyzed the transcription levels of *CsCaM1*, *CsCaM2* and *CsCaM3* in transgenic cucumber cotyledons (Figure 3). In our experiment, the transcript levels of *CsCaM1*, *CsCaM2*, and *CsCaM3* were upregulated in *CsMLO1*-silenced cucumber cotyledons compared with those in the TRV::00 plants, although the transcript level of *CsCaM3* was more discernibly regulated than *CsCaM1* and *CsCaM2*. *CsCaM1* and *CsCaM3* expression was reduced while *CsCaM2* expression was increased in the *CsMLO1*-overexpressing plants compared with that in the control plants. However, only the transcript level of *CsCaM1* was increased in *CsMLO2*-silenced cucumber cotyledons but no alteration in gene expression occurred in the other patterns. Based on the above results, the stronger correlation between *CsMLO1* and *CsCaM3* showed that transient *CsMLO1* expression negatively regulated *CsCaM3* accumulation in cucumber cotyledons. However, the gene expression of *CsCaMs* was not significantly regulated by *CsMLO2*.

### 2.2. CsMLO1 Interacts with CsCaM3 in Yeast and Plants

Next, we examined the interaction between CsMLO1 or CsMLO2 and CsCaM1, CsCaM2, and CsCaM3 (Figure 4). Two cDNAs from the CaM-binding domain of CsMLO1 and CsMLO2 (CaMBD2 and CaMBD1, respectively) were constructed into pGBKT7 (BD/CsMLO1 and BD/CsMLO2). Three cDNAs from CsCaM1, CsCaM2, and CsCaM3 were constructed into pGADT7 (AD/CaM1, AD/CaM2, AD/CaM3, respectively). All yeast strains cotransformed with plasmids were grown on synthetic dropout (SD) medium lacking Trp-Leu-His-Ade. The results showed that the yeast combinations BD/CsMLO1+AD/CsCaM3 grew well on SD/-Trp-Leu-His-Ade plates while other strains cotransformed with plasmids did not grow normally (Figure 4A). Collectively, these results indicated that CaMBD1 of CsMLO1 specifically interacted with CsCaM3 in the yeast two-hybrid system.

The LUC assay is a convenient system and provides a rapid analysis of protein functions in *N. benthamiana* plants presenting *Agrobacterium*-mediated transient expression of genes [29,30]. Therefore, we tested the LUC activity in leaves coexpressing different constructs by *Agrobacterium*-mediated transient expression. Figure 4B shows that nLUC-MLO1 (CaMBD1) and the empty cLUC vector did not display LUC complementation while coinfiltration of *Agrobacterium* containing nLUC-MLO1 (CaMBD1) and cLUC-CaM1/cLUC-CaM2/cLUC-CaM3 resulted in LUC complementation. However, strong fluorescent signaling was detected at nLUC-MLO1 (CaMBD1) and cLUC-CaM3. We also detected LUC complementation following the coexpression of nLUC-MLO2 (CaMBD2) and cLUC-CaM1/cLUC-CaM2/cLUC-CaM3, which showed a very low background as the negative control construct containing nLUC-MLO2 and the empty cLUC vector. Western blotting showed that the corresponding proteins were coexpressed in tobacco leaves, indicating that the LUC fluorescence of tobacco leaves was due to interprotein interactions.

The interaction in plants was also examined via a BiFC analysis [31]. The NH_2_-proximal half of yellow fluorescent protein (YFP) (nYFP) was fused to CsMLO1 and CsMLO2; and the C-proximal half of CFP (cCFP) was fused to CsCaM1, CsCaM2, and CsCaM3. Six combinations of fusion proteins, CsMLO1-nYFP+CsCaM1-cCFP, CsMLO1-nYFP+CsCaM2-cCFP, CsMLO1-nYFP+CsCaM3-cCFP, CsMLO2-nYFP+CsCaM1-cCFP, CsMLO2-nYFP+CsCaM2-cCFP, and CsMLO2-nYFP+CsCaM3-cCFP, were transiently co-overexpressed in *N. benthamiana* leaves (Figure 4C). The results showed that the green fluorescence was increased in leaves transiently co-overexpressing CsMLO1-nYFP+CsCaM1-cCFP, CsMLO1-nYFP+CsCaM2-cCFP, and CsMLO1-nYFP+CsCaM3-cCFP compared with the leaves of the control (CsMLO1-nYFP+cCFP), indicating that CsMLO1 interacted with CsCaM1, CsCaM2, and CsCaM3. Interestingly, the green fluorescence of the CsMLO1-nYFP+CsCaM3-cCFP fusion proteins was brighter than that of the other fusion proteins (CsMLO1-nYFP+CsCaM1-cCFP and CsMLO1-nYFP+CsCaM2-cCFP). However, no or weaker green fluorescence was detected in the fusion proteins (CsMLO2-nYFP+CsCaM1-cCFP, CsMLO2-nYFP+CsCaM2-cCFP, and CsMLO2-nYFP+CsCaM3-cCFP). The above experiments indicate that the interaction between CsMLO1 and CsCaM3 was highly stable. Collectively, our results supported the hypothesis that CsMLO1 interacted with CsCaM3 in plants.

### 2.3. Subcellular Colocalization of CsMLO1 and CsCaM3 in Plants

To investigate whether *CsCaM3* localization was altered via the coexpression with *CsMLO1*, transient coexpression of *CaCaM3* and *CaMLO1* was observed in *N. benthamiana* leaves and protoplasts (Figure 5; Figure 6). *CsCaM3*-GFP fluorescence was predominantly located in the cytosol and plasma membrane, which was the same as in the GFP control, while *CsMLO1*-GFP was mainly localized in the plasma membrane. The coexpression of *CsCaM3*-GFP and *CsMLO1*-GFP transferred the fluorescence of *CsCaM3*-GFP from the cytoplasm to the plasma membrane, thereby resulting in a significant reduction in the fluorescence emission of *CsCaM3*-GFP compared to that in cells expressing *CsCaM3*-GFP alone. However, the coexpression of the empty vector and *CsMLO3*-GFP did not reveal this phenomenon of the fluorescence transfer of *CsMLO3*-GFP (Appendix A). This result further confirmed the interaction between *CsCaM3*-GFP and *CsMLO1*-GFP in the tobacco epidermis and protoplasts. Additionally, *CsMLO1*-GFP blocked the accumulation of *CsCaM3*-GFP in tobacco cells. Here, the result of *CsMLO1* negatively regulating *CsCaM3* expression was further confirmed.

### 2.4. Inhibition of CsCaM3-Triggered Defense Response in CsMLO1-Overexpressing Cucumber Cotyledons

To examine the function of coexpressing *CsMLO1* and *CsCaM3* in the defense response of cucumber plants to *C. cassiicola* infection, *Agrobacterium* carrying the 35S::00, 35S::*CsMLO1*, 35S::*CsCaM3*, and 35S::*CsMLO1*/35S::*CsCaM3* constructs was infiltrated into the cucumber cotyledons (Figure 7). Compared with the control cucumbers, cucumbers transiently overexpressing *CsMLO1* had strongly induced necrotic lesions in the cotyledons at 5 dpi of *C. cassiicola* invasion. However, transient overexpression of *CsCaM3* reduced necrotic lesions in cucumber cotyledons, whereas coexpression of *CsMLO1* and *CsCaM3* in cucumber cotyledons enhanced necrotic lesions compared with the overexpression of *CsCaM3* alone. In addition, we also investigated disease status via a disease index analysis (Table 1). *CsMLO1*-overexpressing cucumber cotyledons were susceptible and presented an infection index of 73.61%, and *CsCaM3*-overexpressing cucumber cotyledons showed high resistance and presented a low infection index of 21.48%. However, the disease index (49.46%) of *CsMLO1*/*CsCaM3*-overexpressing cucumber cotyledons significantly increased or decreased compared with that of *CsCaM3*- or *CsMLO1*-overexpressing cucumber cotyledons, respectively.

To define the defense response of cucumber plants at the cellular level, we performed various histochemical analyses on cucumber cotyledons infected with *C. cassiicola* for 3 days (Figure 8). The induction of HR-like cell death and callose deposition was significantly increased in *CsCaM3*-overexpressing cucumber cotyledons at 5 days after *Agrobacterium* transformation by trypan blue and aniline staining analysis, although the coexpression of *CsMLO1* and *CsCaM3* significantly suppressed cell death in these tissues (Figure 8A). When challenged with *C. cassiicola*, the cucumber cotyledons overexpressing *CsCaM3* demonstrated the highest rate of cell death, while the cucumber cotyledons overexpressing *CsMLO1* demonstrated the lowest rate of cell death. Cell death and callose deposition were reduced in plant tissues coexpressing *CsMLO1* and *CsCaM3* compared to plant tissues overexpressing *CsCaM3*, and this reduction might be mediated by *CsMLO1* after 3 days of *C. cassiicola* infection (Figure 8B). Similarly, the *CsCaM3*-overexpressing cucumber cotyledons showed large brown spots compared with the empty-vector control, and this phenomenon showed that the cotyledons accumulated high levels of H_2_O_2_ in the *CsCaM3*-overexpressing cucumber cotyledons when detected by 3,3′-diaminobenzidine (DAB, a histochemical reagent for H_2_O_2_) staining. Interestingly, coexpression of *CsMLO1* and *CsCaM3* in cucumber cotyledons strongly reduced H_2_O_2_ accumulation compared to that in the *CsCaM3*-overexpressing cucumber cotyledons. Furthermore, the infection of cucumber cotyledons also triggered H_2_O_2_ accumulation (Figure 8A). The accumulation of H_2_O_2_ was evidently increased in the *CsCaM3*-overexpressing cucumber cotyledons after inoculation with *C. cassiicola*, although this accumulation was also inhibited in the *CsMLO1* and *CsCaM3*-coexpressing cucumber cotyledons based on DAB staining (Figure 8B).

Transcription level measurements supported significantly attenuated or strengthened cell death and H_2_O_2_ burst in *CsMLO1*- or *CsCaM3*-overexpressing cucumber cotyledons, respectively (Figure 9; Figure 10). *CsCaM3* overexpression strongly induced H_2_O_2_ signaling-related genes, including ascorbate peroxidase (*CsPO1*) and NADPH oxidase homolog (*CsRbohD* and *CsRbohF*), including defense-related genes (*CsPR1* and *CsPR3*) and callose deposition-related gene (*CsGSL*). However, these related genes were not induced in *CsMLO1*-overexpressing cells (Figure 9). Notably, the extent of the increase in the transcription levels of these genes was significantly inhibited after coexpression of *CsMLO1* and *CsCaM3.* To examine whether the transcription level of related defense genes in transgenic cucumbers was changed under the stress of *C. cassiicola*, we also detected the expression of the above related genes (Figure 10). The data showed that changes in the expression of these genes were consistent with the trend of the above analyzed defense-related genes. Interestingly, coexpression of both *CsMLO1* and *CsCaM3* in cucumber cotyledons strongly inhibited the expression of *CsPR3* compared to that in the *CsCaM3*-overexpressing cotyledons. Collectively, these results suggested that *CsMLO1* overexpression suppressed *CsCaM3*-regulated cell death and defense responses to *C. cassiicola*, including defense gene expression and ROS burst. The above experiments demonstrated that *CsMLO1* can negatively regulate the expression of *CsCaM3*, thereby changing the HR effect of cucumber resistance on *C. cassiicola*.

### 2.5. CsMLO1 Silencing Enhances Cucumber Resistance to C. cassiicola Infection

TRV-based VIGS technology has been effectively used in cucumber reverse genetics research and can detect the functions of defense-related genes in cucumber plants [14,27,28]. The defense resistance of *CsMLO1* silencing was evaluated following *C. cassiicola* inoculation for 5 days in transgenic cucumbers (Figure 11). The *CsMLO1*-silenced cucumbers exhibited enhanced disease resistance as demonstrated by milder symptoms than those of the controls (Figure 11A). To determine whether *CsMLO1* silencing regulated the expression of genes involved in ROS signaling and cell death under *C. cassiicola* infection, we analyzed the transcript levels of these defense-related genes by RT-qPCR (Figure 11B). After 5 days of infection, *CsMLO1* silencing distinctly induced the expression of genes, including *CsPO1*, *CsRhobD*, *CsRhobF*, *CsPR1*, *CsPR3*, *CsGSL*, and *CsCaM3*, compared with the empty-vector control leaves. In particular, the transcript levels of *CsPR3* were more significantly multiplied in *CsMLO1*-silenced cucumber cotyledons. These results indicated that a high level of defense-related genes is required for the *CsMLO1*-mediated resistance response to *C. cassiicola*.

## 3. Discussion

Traditionally, MLO functions have been associated with susceptibility and resistance to powdery mildew disease [32]. Powdery mildew, as an ascomycete pathogen, can infiltrate into epidermal host cells and trigger various plant defense-related responses, including transcriptional activation of *PR* genes, callose deposition of papillae, and biosynthesis of antimicrobial biomolecules [33]. In *Arabidopsis*, the callose deposition in papillae acts as a physical barrier against powdery mildew invasion to improve resistance [18,34]. Callose is a polymeric (1→3)-β-D-glucan that is synthesized by plasma membrane-resident glucan synthase-like (GSL) proteins. In an *Arabidopsis mlo* mutant genetic screen, GSL5-generated callose is mainly deposited in the wound site and pathogen-triggered papillae [18]. The *Arabidopsis mlo2* mutation confers partial resistance to *Golovinomyces orontii*, while the *mlo2 mlo6 mlo12* mutation results in complete immunity. These phenomena are mainly due to the prevention of fungal onset before the fungus successfully penetrates the host cell wall [18]. Papilla formation is usually accompanied by a spatial and temporal burst of ROS, and ROS act as a signal transduction intermediate to activate the relevant signaling pathways [35,36]. Studies in barley are consistent with those in *Arabidopsis* because the *mlo* mutants also enhanced callose deposition, ROS bursts and the pathogen-induced expression of *PR* genes in resistance to *Blumeria graminis* f. sp. *hordei* (*Bgh*) [34]. Therefore, *MLO* can negatively regulate the resistance of powdery mildew in plants.

Additionally, barley and *Arabidopsis mlo* resistance requires three resistance components, i.e., the *PEN* genes, which were initially discovered to defend against nonadaptive powdery mildew *Blumeria graminis* f. sp. *hordei* and *Erysiphe pisi* [9,12]. Inducible defense responses in nonhost plants include the accumulation of ROS, activation of pathogenesis-related genes, localized reinforcement of the plant cell walls and HR [6,37]. Currently, information regarding the defense mechanisms of the cucumber–*C. cassiicola* interaction is limited. In our previous study, an iTRAQ analysis suggest that the occurrence of dynamic changes in the early stages of *C. cassiicola* stress. The accumulation of NADPH oxidases (RBOHs) and level of PR proteins in cucumber leaves under *C. cassiicola* infection at 24 h post-infection show that cucumber plants increase the synthesis of ROS and defense-related genes to prevent the invasion of pathogenic fungi [13]. Wang has shown that secondary metabolism and ROS accumulation play important roles in disease resistance during cucumber–*C. cassiicola* interactions [16]. Moreover, we first found that *CsMLO1* is a negative modulator that regulates the defense response of cucumbers to *C. cassiicola* [15]. Next, the relationships of *CsPR* and *ROS*-associated genes with the overexpression and silencing of *CsMLO1* in noninfiltrated cucumber plants was investigated, and the results show that *CsMLO1* acts as a negative regulator to activate/inhibit the expression of related genes [15]. Thus, we propose that these genes, including ROS-related genes and defense marker genes, are important in cucumber immunity to *C. cassiicola* infection. The function of the HR-related gene *CaM* has been fully studied in recent years. For example, the transient expression of *CaCaM1* increased the ROS burst and hypersensitive cell death to improve the defense responses in pepper [23]. In this study, *CsMLO1* and *CsCaM3* are linked to the defense response to *C. cassiicola,* and we hypothesized that the resistance mechanism of *CsMLO1* and *CsCaM3* in the regulation of cucumbers to *C. cassiicola* was related to NHR.

We adopted the yeast two-hybrid, LUC and BiFC methods to test the interaction and role of CsMLO and CsCaM in plant defense responses. The results showed that CsMLO1 stably interacted with CsCaM3. In barley, a CaM-binding domain (CaMBD) in MLO interacts with Ca^2+^-dependent calmodulin proteins, and this loss of binding halves the ability of MLO to negatively regulate powdery mildew in vivo [20]. Pathogen-triggered Ca^2+^ fluxes can enhance the oxidative burst to activate resistance responses [16,17]. However, increasing Ca^2+^ ions also promotes the affinity of CaM for its target protein, MLO, to suppress resistance in barley [20]. At present, the function of the CaM–MLO complex is rarely studied in cucumber resistance or susceptibility to *C. cassiicola*. In the *Agrobacterium*-mediated transient transformation, *CsCaM3* was transferred from the cytoplasm to the plasma membrane with a reduction in *CsCaM3* accumulation when *CsMLO1* and *CsCaM3* were coexpressed in tobacco cells. Furthermore, *CsMLO1* in transiently transformed transgenic cucumbers can negatively regulate the expression of *CsCaM3* by RT-qPCR. In contrast, *CsMLO1* silencing in cucumber plants significantly enhances the expression of the cell death marker genes *CsCaM3* and *CsPO1*, ROS-related genes, defense marker genes, and callose deposition-related genes during *C. cassiicola* infection. In pepper, the pathogen-response genes *CaCaM1* and *CaPO2* can activate ROS signaling and HRs to improve defense resistance in pepper leaves [23]. *CaMLO2* silencing enhanced resistance against virulent *Xcv* infection, which is related to ROS bursts and *PR* gene expression, suggesting that *CaMLO2* plays a role in abolishing stress-induced cell death and H_2_O_2_ burst at the infection site [38]. Collectively, these findings support the hypothesis that *CsMLO1* can negatively regulate *CsCaM3* expression to affect the HR of cucumbers to *C. cassiicola*.

Further studies are required to define the molecular mechanisms underlying CsCaM3–CsMLO1 complex-induced cell death signaling in the cucumber-*C. cassiicola* interaction. *Agrobacterium*-mediated transient *CsCaM3* overexpression induced cell death and enhanced defense against *C. cassiicola*, suggesting that *CsCaM3* might act as a positive regulator of cell death and disease resistance in cucumber plants. As expected, cell death, callose deposition, and ROS burst were rapidly induced in *CsCaM3*-overexpressing leaves under *C. cassiicola* infection. In pepper, early defense indicators have been detected, such as callose deposition and ROS generation during infection, suggesting that transient *CaCaM1* overexpression induces localized cell death and enhances locally acquired resistance against virulent Xcv in pepper leaves [38,39]. Furthermore, coexpression of *CaCaM1* and *CaMLO2* significantly suppresses *AvrB*-triggered cell death and defense responses [40]. In our study, coexpression of *CsMLO1* with *CsCaM3* also significantly reduced HR-associated cell death in response to *C. cassiicola* infection. Increasing evidence indicates that plant responses to pathogen invasions via the regulation of the accumulation of ROS, such as H_2_O_2_ and O_2_·-, are related to *AtrbohD* and *AtrbohF* [41,42,43,44,45,46]. ROS generation and cell death induction are also regulated by the expression of the peroxidase gene *CaPO2* in pepper leaves [47]. Under *C. cassiicola* infection, the mRNA levels of *CsPO1*, *CsRbohD*, and *CsRbohF* were significantly increased in *CsCaM3*-overexpressing cotyledons, although their transcription levels were inhibited via the coexpression of *CsCaM3* and *CsMLO1* in cucumber. Furthermore, triggering HR-like cell death is accompanied by callose deposition and defense marker gene induction, including pathogenesis-related protein 1-1a (*PR1-1a*) and chitinase (*PR3*), which are upregulated to promote resistance to pathogens [18,39,48,49]. Similar to these findings, the strong induction of defense marker genes (*CsPR1* and *CsPR3*) and callose deposition-related genes (*CsGSL*) was observed in *CsCaM3*-overexpressing cotyledons, although the coexpression of *CsCaM3* with *CsMLO1* compromised the accumulation of the transcription levels of *CsPR1* and *CsPR3* and *CsGSL*. Collectively, our findings combined with these data support the hypothesis that the coexpression of *CsMLO1* with *CsCaM3* negatively regulates an early step in *C. cassiicola*-triggered cell death and immunity in cucumber leaves. In our study, we did not detect the interaction of the CsCaM protein with CsMLO2. We speculate that there may be other proteins that interact with CsMLO2, such as the membrane protein ROP (Rho-related GTPase of plant) [13,50]. In addition, the role of *CsMLO2* as a negative regulator in cucumber disease resistance remains unknown. These functional speculations about CsMLO2 will be the focus of further studies.

In conclusion, *CsMLO1* is a negative modulator that enhances the defense response of cucumbers and stably interacts with CsCaM3 and transfers CsCaM3 in the cytoplasm to the plasma membrane, thereby blocking the accumulation of CsCaM3. Furthermore, the coexpression of *CsCaM3* and *CsMLO1* significantly inhibited hypersensitive cell death after *C. cassiicola* infection, suggesting that *CsMLO1* negatively regulates *CsCaM3* expression, which results in the inhibition of defense-related gene activation.

## 4. Materials and methods

### 4.1. Plant Materials and Pathogen Inoculation

Cucumber (*C. sativus* L. cv. Xintaimici) and tobacco (*N. benthamiana*) plants were sown in pots with peat:vermiculite (1:2, *v*/*v*) under greenhouse conditions (25 °C, 16 h light/8 h dark cycle). At the second-third leaf stage, the plants were sprayed with a *C. cassiicola* spore suspension, for which the spores had been harvested in sterile water and adjusted to a final concentration of 10^5^ microconidia/mL. Cucumber leaves were incubated at 25 °C after *C. cassiicola* inoculation, and then the second leaves were collected from three plants at 3 h, 6 h, 12 h, 24 h, 48 h, 72 h, and 144 h. The control plants were inoculated with distilled water. Infected leaves were sampled at various time points for the fungal growth assay, RNA isolation, and the histochemical assay.

The disease progression of corynespora leaf spot was estimated on the basis of the severity of leaf scabs as follows: Grade 0: no scabs observed; Grade 1: less than 1/10 of leaves infected; Grade 3: 1/10–1/4 of leaves infected; Grade 5: 1/4–1/2 of leaves infected; Grade 7: 1/2–3/4 of leaves infected; and Grade 9: more than 3/4 of leaves infected. Disease index = 100 × ∑(no. of diseased leaves of each grade × disease grade) / (total no. leaves × 9).

### 4.2. RNA Extraction, cDNA Synthesis, and Reverse-Transcription Quantitative PCR

Total RNA extraction and cDNA synthesis were performed as previously described [15]. The primers for candidate genes were designed using QuantPrime—a flexible primer design tool for high-throughput qPCR by http://quantprime.mpimp-golm.mpg.de/. RT-qPCR was performed on a SYBR Green I 96-I system (Roche fluorescence quantitative PCR instrument, Basel). The cucumber actin gene was used as the internal reference [51]. The relative expression of the target genes was calculated using the 2^−ΔΔCT^ method [52]. Leaves/cotyledons sampled at each sampling point were divided into three groups (three leaves per group) and then evenly mixed for RNA extraction. RNA extracted from each group was used as one biological replicate for RT-qPCR. A total of three biological replicates were performed. All primers used to detect gene expression are listed in Appendix A.

### 4.3. Yeast Two-Hybrid Assay

The PCR-amplified full-length coding sequence (CDS) regions of *CsCaM1*, *CsCaM2*, and *CsCaM3* were ligated into the pGADT7 vector using the EcoRI and BamHI restriction sites. CaMBD1 (CsMLO1) and CaMBD2 (CsMLO2) were ligated into the pGBKT7 vector using the EcoRI and BamHI restriction sites. The recombinant plasmids BD/CsMLO1, BD/CsMLO2, AD/CsCaM1, AD/CsCaM2, and AD/CsCaM3 were generated. The primers used are shown in Appendix A. BD and AD vectors were cotransformed into the Y2HGold yeast strain. Transformants were arrayed on interaction-selection medium (SD medium lacking adenine, histidine, leucine, and tryptophan, supplemented with 10 mmol·L^−1^ CaCl_2_), and growth was scored as an indicator of the interactions between two proteins.

### 4.4. Firefly Luciferase Complementation Imaging Assay and Western Blot

The CaM-binding domains CaMBD1 (CsMLO1) and CaMBD2 (CsMLO2) were ligated into the pCAMBIA1300-NLUC vector using the KpnI and SalI restriction sites. The CDS regions of CsCaM1, CsCaM2 and CsCaM3 were ligated into the pCAMBIA1300-CLUC vector using the KpnI and SalI restriction sites. After sequencing, 35S::CaMBD1-nLUC, 35S::CaMBD2-nLUC, 35S::cLUC-CaM1, 35S::cLUC-CaM2, and 35S::cLUC-CaM3 fusion constructs were transformed into *A. tumefaciens* strain EHA105. *Agrobacterium* suspensions harboring the 35S::cLUC vector/35S::CaMBD1-nLUC, 35S::cLUC-CaM1/35S::CaMBD1-nLUC, 35S::cLUC-CaM2/35S::CaMBD1-nLUC, 35S::cLUC-CaM3/35S::CaMBD1-nLUC, 35S::cLUC vector/35S::CaMBD2-nLUC, 35S::cLUC-CaM1/35S::CaMBD2-nLUC, 35S::cLUC-CaM2/35S::CaMBD2-nLUC, and 35S::cLUC-CaM3/35S::CaMBD2-nLUC were infiltrated into *N. benthamiana* leaves using a needleless syringe. Luciferase activity was detected 3 days after infiltration using the 699 NightSHADE LB 985 imaging system (Berthold Technologies). Then, 0.2 mmol·L^−1^ luciferin (Promega; Madison, WI, USA) was sprayed on the surface of the tobacco leaves, and the LUC fluorescence intensity was measured after 5 min. 

Total proteins were extracted from infiltrated leaves by grinding 0.3 g of leaf tissue in 3 mL buffer (0.0625 mol·L^−1^ Tris-HCl, pH 6.8, 0.5% SDS, 5% glycerin, 3% β-mercaptoethanol and 100% Triton X-100) at 4 °C for 1 h. Then, the solution was centrifuged at 10,000× *g* for 20 min at 4 °C. The resulting supernatant was the total protein extract fraction. The supernatant was supplemented with 10 mmol·L^−1^ CaCl_2_ and four volumes of acetone solution at -20 °C overnight. The next day, the crude extract was clarified by centrifugation at 10,000× *g* for 20 min at 4 °C. Twenty-four milligrams of protein dry powder was added to 50 µL of protein sample treatment solution (0.125 mol·L^−1^ Tris-HCl, pH 8.0, 2% SDS, 5% β-mercaptoethanol, 10% glycerin and 0.02% bromophenol blue) and boiling water for 3–5 min and then stored at −20 °C. The fractionated proteins were subjected to SDS-PAGE, and immunoblotting was performed using standard methods. Immunoprecipitates were electrophoretically separated by SDS-PAGE and transferred to a nitrocellulose membrane (Amersham, Shanghai, China). Related proteins (Flag-tagged CsCaMBD1, Flag-tagged CsCaMBD2, HA-tagged CsCaM1, HA-tagged CsCaM2 and HA-tagged CsCaM3 using Flag and HA antibodies) were detected by immunoblot using antibodies anti-Flag (Beyotime, Shanghai, China) and anti-HA (Beyotime, Shanghai, China). Anti-Flag and anti-HA were diluted 1:1000 with TBST buffer. The secondary antibody (anti-mouse, Cell Signaling Technology, Boston, MA, USA) was diluted 1:1000 with TBST buffer. The primers used are shown in Appendix A.

### 4.5. Bimolecular Fluorescence Complementation Analysis 

A BiFC analysis was performed to identify the interactions between CsMLO1/CsCaM1, CsMLO1/CsCaM2, CsMLO1/CsCaM3, CsMLO2/CsCaM1, CsMLO2/CsCaM2, and CsMLO2/CsCaM3. The BiFC constructs with the full-length CDS regions CsMLO1, CsMLO2, CsCaM1, CsCaM2, and CsCaM3 (without termination codons) were ligated into intermediate vectors pDONR221 by the BP reaction. Finally, the target gene fragments were individually constructed into the corresponding functional vectors PXNGW (nYFP) and PXCGW (cCFP) by the LR reaction. *A. tumefaciens* strain GV3101 containing BiFC constructs was coinfiltrated into *N. benthamiana* leaves. Leaves were visualized 2–3 days after *agroinfiltration* using a confocal laser scanning microscope (Leica TCS SP8, Solms, Germany) to detect YFP. All assays were repeated independently at least three times with comparable results. The primers used are shown in Appendix A.

### 4.6. Construction of Fusion Proteins, Plant Transformation, and Fluorescence Microscopy

The CDSs of *CsMLO1* and *CsCaM3* were inserted between the CaMV 35S promoter and the soluble-modified GFP (eGFP) in the pRI101-GFP vector to generate the eGFP-fusion proteins *CsMLO1*-GFP and *CsCaM3*-GFP, respectively. To determine *CsCaM3* localization in the presence of *CsMLO1*, the *CsCaM3*-GFP construct was cobombarded with pRI10135S::*CsMLO1* lacking the C-terminal GFP. Recombinant plasmids were transformed into *A. tumefaciens* strain EHA 105 and then centrifuged after overnight culture. The precipitate was cultured in induction medium (10 mmol·L^−1^ ethanesulfonic acid (pH 5.7), 10 mmol·L^−1^ MgCl_2_ and 200 mmol·L^−1^ acetosyringone), harvested, diluted to OD_600_ = 0.6, and then injected into *N. benthamiana* leaves. Two days after infiltration, the epidermal cells and protoplasts were observed using a confocal laser scanning microscope (Leica TCS SP8, Solms, Germany). The filter sets BP505-530 (excitation 488 nm, emission 505 to 530 nm) were used to detect GFP. Protoplasts were extracted from transformed tobacco leaves, which were cut into 0.5–1 mm thin strips, quickly transferred to the enzymatic hydrolysate (20 mmol·L^−1^ MES, 1.5% Cellulase R-10, 0.4% Macerozyme R-10, 20 mmol·L^−1^ KCL, 0.4 mol·L^−1^ mannitol, 10 mmol·L^−1^ CaCl_2_, and 0.1% BSA, pH 5.7), subjected to 30 min of vacuum infiltration, and then shaken at 50 rpm for 6 h. The enzymatic hydrolysate was diluted with an equal amount of W5 (0.2 mol·L^−1^ MES, 1.54 mol·L^−1^ NaCl, 1 mol·L^−1^ CaCl_2_, and 0.2 mol·L^−1^ KCl, pH 5.7) and then filtered through a 200-m nylon membrane. The collected filtrate was centrifuged at 100× *g* for 2 min. The supernatant was slowly removed, and the remaining green liquid contained protoplasts. An equal amount of precooled MGG was placed in the protoplasts, which were placed on ice for observation using a confocal laser scanning microscope. The primers used are shown in Appendix A.

The CDSs of *CsMLO1* and *CsCaM3* were inserted between the CaMV 35S promoter and the soluble-modified green fluorescent protein (GFP) (eGFP) in the pRI101-GFP vector to generate the eGFP-fusion proteins *CsMLO1*-GFP and *CsCaM3*-GFP, respectively. To determine *CsCaM3* localization in the presence of *CsMLO1*, the *CsCaM3*-GFP construct was cobombarded with pRI10135S::*CsMLO1* lacking the C-terminal GFP. Recombinant plasmids were transformed into *A. tumefaciens* strain EHA 105, which were centrifuged after overnight culture. The precipitate was cultured in induction medium (10 mmol·L^−1^ ethanesulfonic acid, pH 5.7, 10 mmol·L^−1^ MgCl_2_ and 200 mmol·L^−1^ acetosyringone), harvested and diluted to OD_600_ = 0.6, and then injected into *N. benthamiana* leaves. Two days after infiltration, the epidermal cells and protoplasts were observed using a confocal laser scanning microscope (Leica TCS SP8, Solms, Germany). The filter sets BP505-530 (excitation 488 nm, emission 505 to 530 nm) were used to detect GFP. Protoplasts were extracted from transformed tobacco leaves. The transformed tobacco leaves were cut into 0.5–1 mm thin strips, quickly transferred to the enzymatic hydrolysate (20 mmol·L^−1^ MES, 1.5% Cellulase R-10, 0.4% Macerozyme R-10, 20 mmol·L^−1^ KCL, 0.4 mol·L^−1^ mannitol, 10 mmol·L^−1^ CaCl_2_, and 0.1% BSA, pH 5.7), and after 30 min of vacuum infiltration, shaken at 50 rpm for 6 h. The enzymatic hydrolysate was diluted with an equal amount of W5 (0.2 mol·L^−1^ MES, 1.54 mol·L^−1^ NaCl, 1 mol·L^−1^ CaCl_2_, and 0.2 mol·L^−1^ KCl, pH 5.7) and then filtered through a 100 μm nylon membrane. The collected filtrate was centrifuged at 100× *g* for 2 min. The supernatant was slowly removed, and the remaining green liquid contained protoplasts. An equal amount of precooled MGG was placed in the protoplasts and placed on ice for observation using a confocal laser scanning microscope. The primers used are shown in Appendix A.

### 4.7. Histochemical Analysis

For the visualization of H_2_O_2_ accumulation with 3,3’-diaminobenzidine (DAB; Sigma, St. Louis, MO, USA) [53], cucumber cotyledons transiently expressing *CsMLO1* and/or *CsCaM3* were assessed by DAB staining. The infected cucumber cotyledons were soaked in 1 mg·mL^−1^ DAB for 8 h, boiled for 20 min in 3:1:1 ethanol/lactic acid/glycerol and then transferred to 95% ethanol at 4 °C for storage. Three independent replicates were performed for each assay.

We detected the cell death of cucumber cotyledons by trypan blue staining [54]. The treated cucumber cotyledons were immersed in trypan blue solution, boiled for 10 min, and placed at 37 °C overnight. Then, the dyed leaves were repeatedly subjected to decolorizing and photographed.

We detected callose deposition in cucumber cotyledons transiently expressing *CsMLO1* and/or *CsCaM3* by aniline blue staining [55,56]. The cucumber cotyledons were stored in FAA fixative (50% ethanol, 5% glacial acetic acid, 10% formaldehyde and 35% H_2_O) for 24 h. The cotyledons were carefully washed with 100% ethanol, and 500 μL of 50% ethanol was added and incubated for 30 min on the bench after removing 100% ethanol. After ethanol was removed, 500 μL 67 mM K_2_HPO_4_ (pH 12) was added, and the leaves were incubated for 30 min. Finally, 500 μL staining solution (0.01% aniline blue in 67 mmol·L^−1^ K_2_HPO_4_) was added, and the cotyledons were incubated for 1 h in the dark to perform UV microscopy. Three independent replicates were performed for each assay.

### 4.8. Cucumber Cotyledon Transformation

The CDS regions of *CsMLO1* and *CsMLO2* were ligated into the pRI101-eGFP vector using the SalI and BamHI restriction sites as previously described. Then, recombinant plasmids were generated to transiently overexpress the *CsMLO1* gene and *CsMLO2* gene in cucumber cotyledons. pTRV-based VIGS was performed to knock down the *CsMLO1* gene and *CsMLO2* gene in cucumber cotyledons. A 402 bp fragment within the 3′ region of *CsMLO1* cDNA (nucleotides 1348–1749) and a 387 bp fragment within the 3′ region of *CsMLO2* cDNA (nucleotides 1347–1725) were cloned into the pTRV vector (TRV:*CsMLO1* and TRV:*CsMLO2*, respectively) as previously described [16]. All recombinant plasmids were transformed into *A. tumefaciens* strain EHA105. *Agrobacterium*-mediated transformations with the 35s::GFP, GFP::*CsMLO1*, GFP::*CsMLO2*, pTRV, pTRV-*CsMLO1*, and pTRV-*CsMLO2* genes were carried out. The samples were supplemented with 10 mmol·L^−1^ MES, 10 mmol·L^−1^ MgCl_2_, and 200 μmol·L^−1^ AS and then infiltrated into fully expanded cotyledons of cucumber plants (OD_600_ = 0.4 for each construct). The plants were then placed in a growth room at 22 °C with a 16 h light and 8 h dark photoperiod for growth.

### 4.9. Statistical Analysis

Primer design and sequence alignment were conducted using Primer 5 software. Data are the mean ± standard deviation from three biological replicates per cultivar. Standard errors of deviation were assessed by Excel. Statistical significance was analyzed by Student’s *t*-test (*P* < 0.05 or *P* < 0.01) using SPSS software.

## Figures and Tables

**Figure 1 ijms-20-02995-f001:**
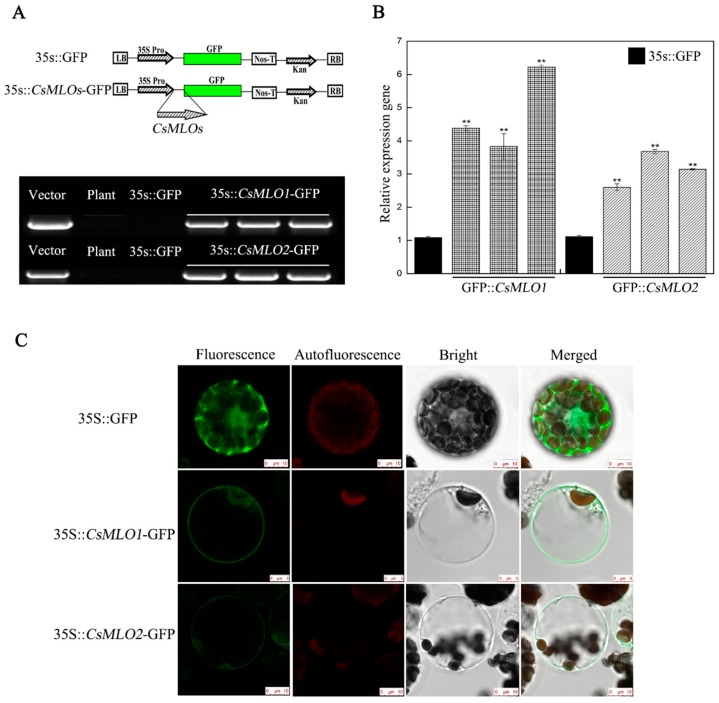
*CsMLO1*- and *CsMLO2*-overexpressing constructs in cucumber cotyledons. (**A**) Schematic of the *CsMLO1-GFP* and *CsMLO2-GFP* constructs. PCR identification of *CsMLO1* and *CsMLO2* genetically modified of cucumber was successful. Vector, recombinant plasmid; Plant, nontransgenic cucumber; 35S::GFP, empty vector; 35S::*CsMLO1*-GFP, *CsMLO1*-overexpressing in cucumbers; 35S::*CsMLO2*-GFP, *CsMLO2*-overexpressing in cucumbers; (**B**) Transgenic plants were identified by RT-qPCR. Data are the means ± standard deviations from three independent experiments, and each column represents a sample containing three cucumber cotyledons from different plants. Expression analysis of candidate genes using the 2^−ΔΔCt^ method. The asterisks indicate a significant difference (Student’s *t* test, ** *P* < 0.01). (**C**) Green fluorescence protein (GFP) was detected in the cucumber protoplasts of the 35S::GFP, 35S::*CsMLO1*-GFP and 35S::*CsMLO2*-GFP constructs. Transient expression in the protoplasts of cucumber cells of 35S::*CsMLO1*-GFP and 35S::*CsMLO2*-GFP were primarily localized in the plasma membrane. Fluorescence, chloroplast autofluorescence, bright field and merged images were obtained using a Leica confocal microscope. Scale bars = 10 μm.

**Figure 2 ijms-20-02995-f002:**
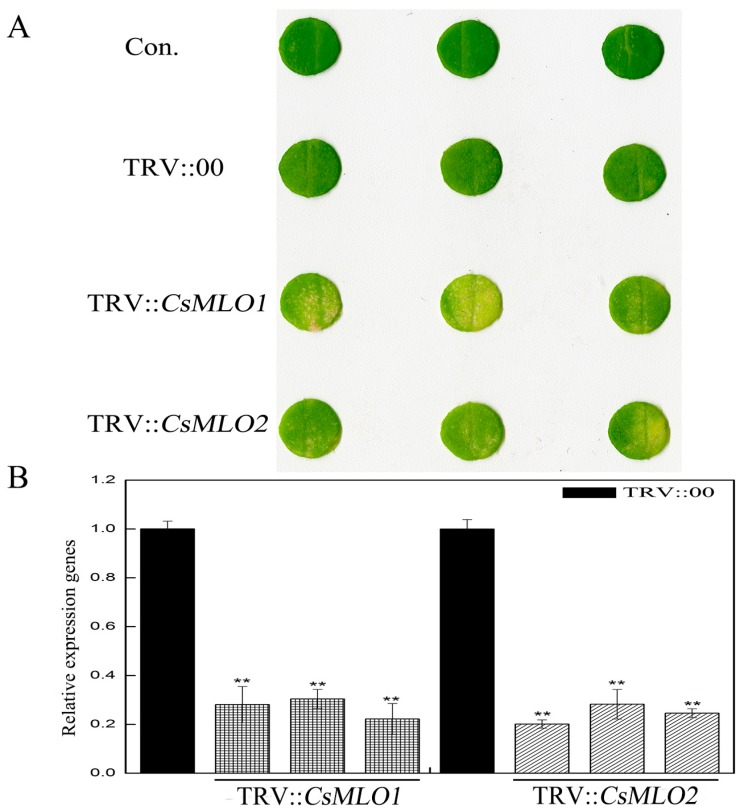
*CsMLO1*-silencing and *CsMLO2*-silencing constructs in cucumber cotyledons. (**A**) Cucumber cotyledons showed chlorotic mosaic symptoms of TRV in *CsMLO1*-silencing and *CsMLO2*-silencing constructs; (**B**) *CsMLO1*-silencing and *CsMLO2*-silencing constructs identified in transgenic plants by RT-qPCR. Black pillars represent the empty vector control; gray grid pillars represent the efficient silencing of *CsMLO1* and *CsMLO2* by RT-qPCR. Data are the means ± standard deviations from three biological experiments, and each column represents a sample containing three cucumber cotyledons from different plants. Expression analysis of the candidate genes using the 2^−ΔΔCt^ method. The asterisks indicate a significant difference (Student’s *t* test, ** *P* < 0.01).

**Figure 3 ijms-20-02995-f003:**
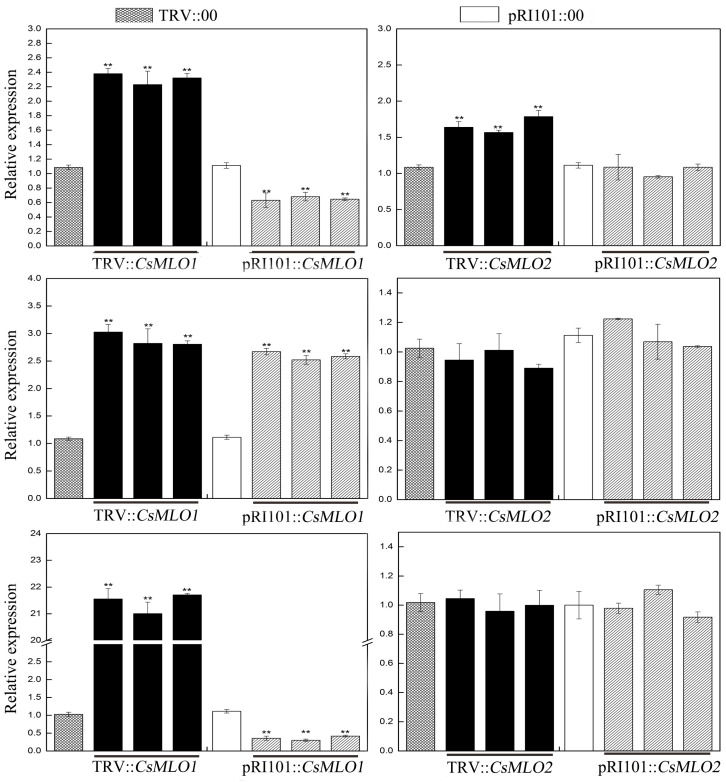
*CsMLO1*-mediated expression patterns of *CsCaM* in cucumber cotyledons via RT-qPCR. Data are the means ± standard deviations from three biological experiments, and each column represents a sample containing three cucumber cotyledons from different plants. Expression analysis of the candidate genes using the 2^−ΔΔCt^ method. The asterisks indicate a significant difference (Student’s *t* test, ** *P* < 0.01).

**Figure 4 ijms-20-02995-f004:**
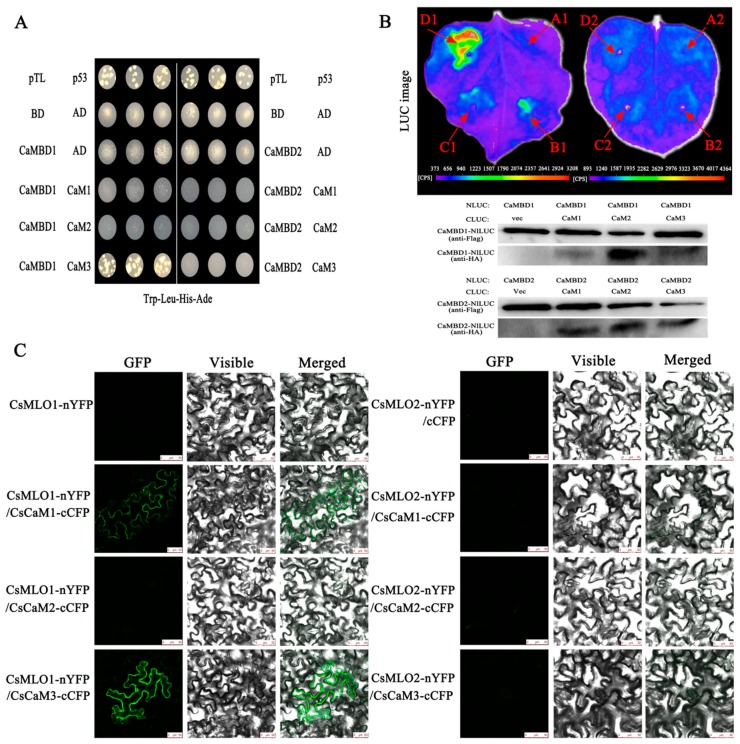
CsMLO interacts with CsCaM in yeast and *N. benthamiana* epidermal cells. (**A**) BD/CsMLO1 or BD/CsMLO2 interacts with AD/CaM1, AD/CaM2, and AD/CaM3 on synthetic dropout medium lacking Trp-Leu-His-Ade in the yeast two-hybrid system. BD, bait; AD, prey; and p53/pTL, positive control. (**B**) Firefly luciferase complementation (LUC) imaging assay and western blot analysis of the interaction between CsMLO and CsCaM in *N. benthamiana* leaves. Arrows indicate leaf panels that were infiltrated with *Agrobacterium* containing the indicated constructs. A1: cLUC+CaMBD1-nLUC; B1: cLUC-CaM1+CaMBD1-nLUC; C1: cLUC-CaM2+CaMBD1-nLUC; D1: cLUC-CaM3+CaMBD1-nLUC; A2: cLUC+CaMBD2-nLUC, B2: cLUC-CaM1+CaMBD2-nLUC, C2: cLUC-CaM2+CaMBD2-nLUC; and D2: cLUC-CaM3+CaMBD2-nLUC. The western blot below shows the expression levels of cLUC- and nLUC-fusion proteins. (**C**) Bimolecular fluorescence complementation (BiFC) analysis of the interaction with CsMLO1/CsCaM1, CsMLO1/CsCaM2, CsMLO1/CsCaM3, CsMLO2/CsCaM1, CsMLO2/CsCaM2, and CsMLO2/CsCaM3 in *N. benthamiana* leaves infiltrated with *Agrobacterium*. (Left) Reconstitution of YFP-derived fluorescence. (Middle) Bright field images. (Right) Merged fluorescence images. (Scale bar: 50 μm.).

**Figure 5 ijms-20-02995-f005:**
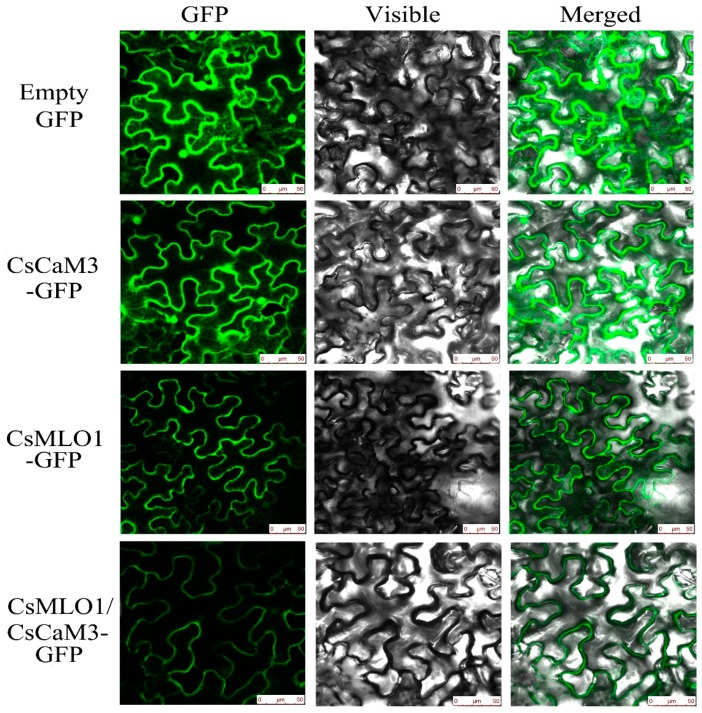
Subcellular localization analyses of *CsCaM3*-GFP, *CsMLO1*-GFP, and *CsCaM3*-GFP+*CsMLO1*-GFP in transiently transformed *N. benthamiana* leaves. Confocal images of green fluorescent protein (GFP), *CsCaM3*-GFP, *CsMLO1*-GFP, and *CsCaM3*-GFP+ *CsMLO1*-GFP in *N. benthamiana* epidermal cells 3 days after *Agroinfiltration*. Bars = 50 μm.

**Figure 6 ijms-20-02995-f006:**
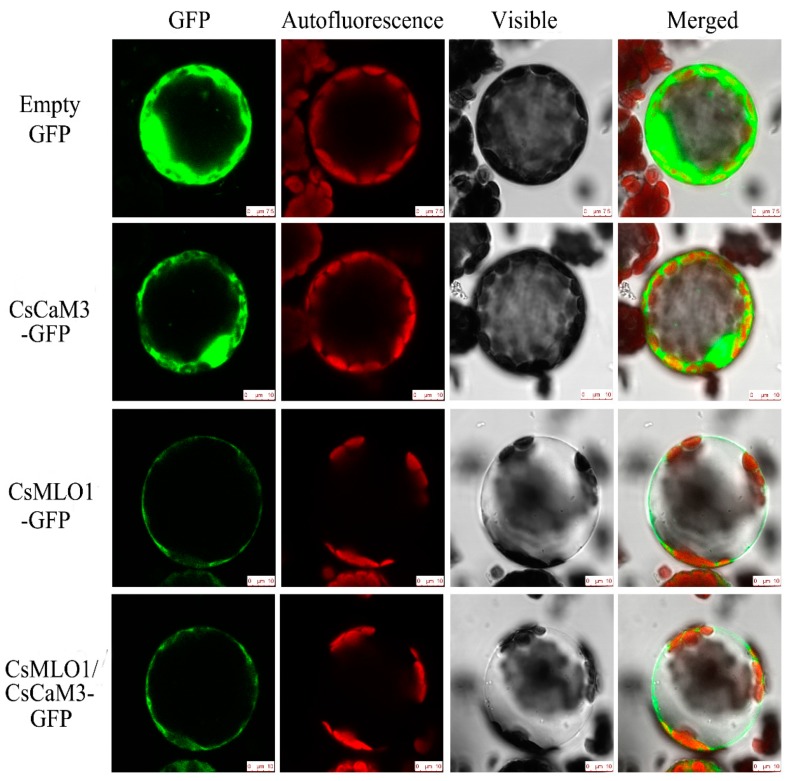
Subcellular localization analyses of *CsCaM3*-GFP, *CsMLO1*-GFP, and *CsCaM3*-GFP+*CsMLO1*-GFP in transiently transformed *N. benthamiana* protoplasts. Confocal images of green fluorescent protein (GFP), *CsCaM3*-GFP, *CsMLO1*-GFP, and *CsCaM3*-GFP+*CsMLO1*-GFP in *N. benthamiana* protoplasts 3 days after *Agroinfiltration*. Chloroplast autofluorescence (red). Bars = 50 μm.

**Figure 7 ijms-20-02995-f007:**
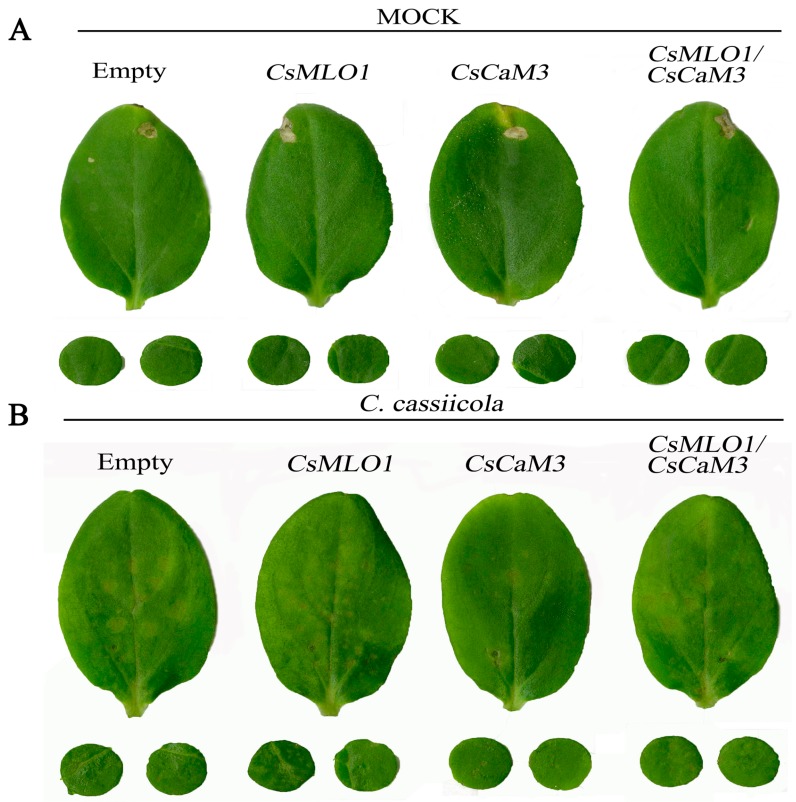
Identification of disease-resistance of empty vector control, *CsCaM3*-overexpressing, *CsMLO1*-overexpressing, and *CsCaM3*-GFP+*CsMLO1*-GFP-overexpressing constructs in cucumber cotyledons. (**A**) Phenotype of uninfected cucumber cotyledons. (**B**) Phenotype of cucumber cotyledons after *C. cassiicola* challenge.

**Figure 8 ijms-20-02995-f008:**
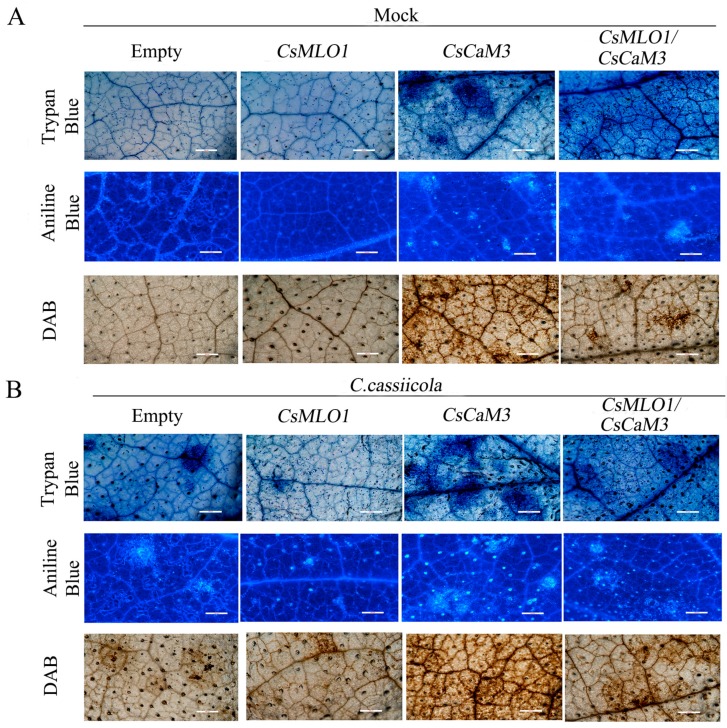
Effects of agroexpression of *CsCaM3* and *CsMLO1* on cell death and defense responses of cucumber leaves. (**A**) Trypan blue, aniline blue, and DAB staining of cucumber cotyledons 5 d after agroexpression with the indicated transgenes. (**B**) Trypan blue, aniline blue, and DAB staining of cucumber cotyledons 3 d after *C. cassiicola* challenge with transgenes. Bars = 500 μm.

**Figure 9 ijms-20-02995-f009:**
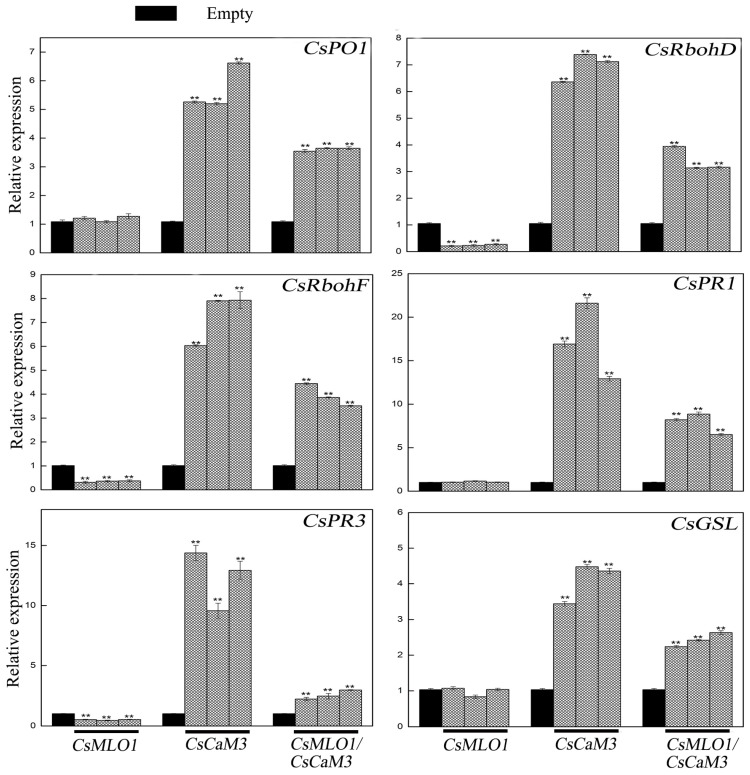
Expression analysis of the *CsRbohD* (cucumber NADPH oxidase homolog D), *CsRbohF* (cucumber NADPH oxidase homolog F), *CsPO1* (cucumber ascorbate peroxidase), *CsPR1* (cucumber defense marker gene), *CsPR3* (cucumber defense marker gene), and *CsGSL* (cucumber callose deposition-related gene) genes were determined by RT-qPCR in cucumber cotyledons 5 d after agroexpression. Data are the means ± standard deviations from three biological experiments, and each column represents a sample containing three cucumber cotyledons from different plants. Expression analysis of candidate genes using the 2^−ΔΔCt^ method. The asterisks indicate a significant difference (Student’s *t* test, ** *P* < 0.01).

**Figure 10 ijms-20-02995-f010:**
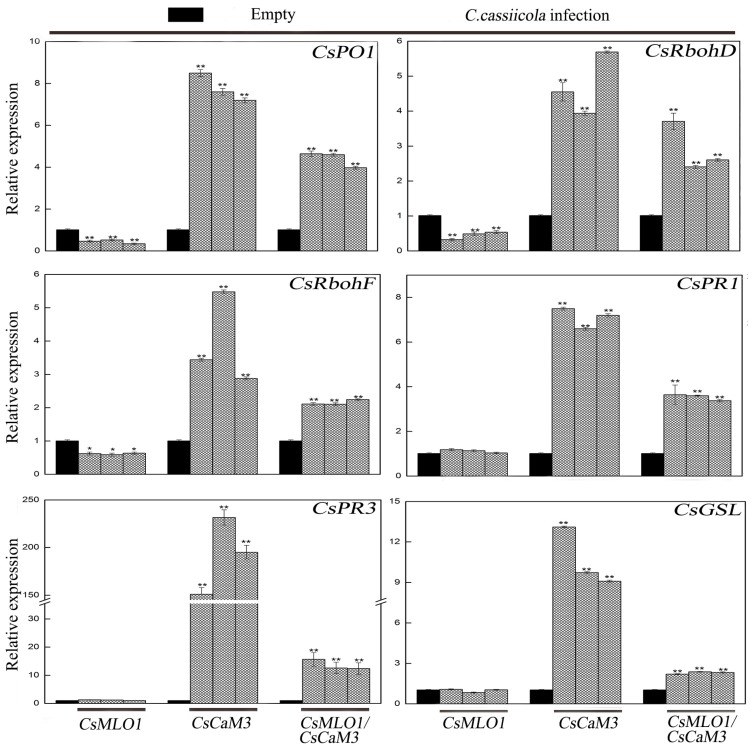
Expression analysis of the *CsRbohD*, *CsRbohF*, *CsPO1*, *CsPR1*, *CsPR3* and *CsGSL* genes were determined by RT-qPCR in cucumber cotyledons 5 d after *C. cassiicola* inoculation. Data are the means ± standard deviations from three biological experiments, and each column represents a sample containing three of cucumber cotyledons from different plants. Expression analysis of candidate genes using the 2^−ΔΔCt^ method. The asterisks indicate a significant difference (Student’s *t* test, * *P* < 0.05 or ** *P* < 0.01).

**Figure 11 ijms-20-02995-f011:**
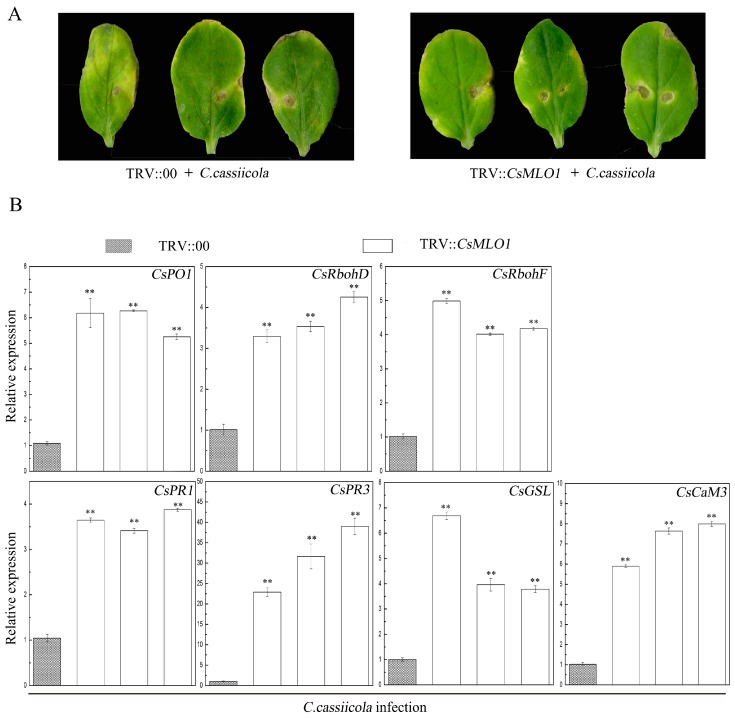
Identification of disease resistance in the empty vector control and *CsCsMLO1* silencing in cucumber cotyledons. (**A**) Phenotype of the cucumber cotyledons after *C. cassiicola* challenge. (**B**) Expression analysis of the *CsRbohD*, *CsRbohF*, *CsPO1*, *CsPR1*, *CsPR3*, *CsGSL* and *CsCaM3* (cucumber Calmodulin) genes were determined via RT-qPCR in *CsMLO1*-silencing cucumber cotyledons 5 d after *C. cassiicola* inoculation. Data are the means ± standard deviations from three biological experiments, and each column represents a sample containing three cucumber cotyledons from different plants. Expression analysis of candidate genes using the 2^−ΔΔCt^ method. The asterisks indicate a significant difference (Student’s *t* test, ** *P* < 0.01).

**Table 1 ijms-20-02995-t001:** Disease index of different cucumber cotyledons to *C. cassiicola*.

Material Name	Disease Index	Resistance
35S::GFP	43.33	S
35S::CsMLO1	73.61	HS
35S::CsCaM3	21.48	MR
35S::CsMLO1/35S::CsCaM3	49.46	S

Note: Note: High resistance (HR), 0 < DI ≤ 15; Moderate resistance (R), 15 < DI ≤ 35; Resistance (MR), 35 < DI ≤ 55; Susceptible (S), 55 < DI ≤ 75; and Highly susceptible (HS), DI > 75. (Wang 2018).

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
