# Peer review of "Cucumber Mildew Resistance Locus O Interacts with Calmodulin and Regulates Plant Cell Death Associated with Plant Immunity"

_ijms, 2019, doi:10.3390/ijms20122995_

Round 1
Reviewer 1 Report
Authors in manuscript entiled: „Cucumber mildew resistance locus O interacts with calmodulin and regulates plant cell death associated with plant immunity” presented analyses of interaction Cucumber mildew resistance locus O - CsMLO1 with calmodulin CsCaM3 associated with reactive oxygen species burst and defense-related gene expression in condition of Corynespora cassicola infection. Authors presented a lot of results, also very interesting conclusions for plant-Ascomycetes interaction, but unfortunately the result are not enough described, not well presented and requires changes and a deep improvement.
Thereofe, the specific comments and suggestion of changes were presented below:
· First of all, did the whole Laboratory the last author of the manuscript, moreover in a role of corresponding author ? please complete the author list’
· Please explain the phrases, which were mentioned in the text first time: KWT3 (calmodulin is not enough), MS/MS analysis
· In results – between line 7-11 failed the citation of the figure or supplement
· Results, line 17 – authors declare citation figure 1 ,which is presented in the manuscript, but simultaneously add to them 3 citation - there is a problem, about which I have reservations. These type of presenting data’s suggest that figure was cited from previous paper , for example, and it is not an original result, so please adapt the figure;
· Figure 1 – confocal microscopy of the GFP-constructs presented in panel C –in panel 35S-GFP and 35S=CsMLO1-GFP authors have signal in cytoplasm and plasmalema, they did not notice the nucleus signal. Here, authors did not have a evidence [in this panel] of membranes, the resolution is definitely too small for that kind of conclusion.
· Please explain in the materials and methods the method of measure the disease index, because there is no information about it
· The authors have a real problem with distinction Cucumber cotyledons from leaves – for example in figure 11 (A) there are for sure the cotyledons ! not leaves! Authors stated that they described and comprised identification of CsMLO1-silencing effect on expression different defense-related genes 5days after treatment – The question is did the authors make the expression in cotyledons or at last in leaves, and the second question is did the authors realized that these expression in cotyledons and leaves could be completely different?
· Figure 3 and 11 it should be comparable scale of relative expression, then the focus on the differences will be more readable, at the moment it looks like rescale charts, unfortunately;
· Figure 4 the LUC image is completely unreadable, it is important and interesting analysis, therefore the micrograph should be enlarged; the same situation with figure 4 C;
· Analysis from figure 9 – what should be the reason of avery high differences between different samples plants in the same “treatment-condition”-especially in CsCaM3 in PO1, PR1 and PR3 ?? Moreover, please ensure that the asterisks of significance are visible;
· The general question is, Why authors choose exactly these ROS related genes to analysis like PO1, RbohD, RbohF ? Reviewer believe that these fungus induced ROS and PR, but why exactly these genes are important in immunity to Corynespora ? there is no information about it, before the mentioned in discussion line 14
· Figure 7- once more there are cotyledons not cucumber leaves ! Moreover, authors used A nad B in the figure legend, but there is not information about it in the figure (??)
· Analyses in figure 8 – authors used statements – evidently increased ,or evidently decreased, but the visualization in UV and light microscopy did not provide these kind of information, authors did not provides quantitative analysis of localization. Photographs in figure 8 are too expanded to conduct observation. Furthermore, there were visible no differences in three localization between transgens (A) and infected C. cassicola (B) – for example between CsMLO1 mock & infected or/and CsMLO1/CsCaM3 mock & infected. So, data about decreasing and increasing based on this figure are too speculative;
· Authors cited your own paper [citation 15] , but they not provided neither vol. nor pages or even the year of the publication;
· Composition of the manuscript, there is a problem, about which it has reservations. We have a lot of presented data –results page 2 to 15 and only 1.5 page of the discussion, which is shorter than introduction.
Therefore, taking into account all these statements above, the reviewer suggest a major revision of the manuscript especially in composition and presentation of the results
Author Response
Firstly, the issue of the article English language has been revisited through the Professional English Language Editing Company.
Q1: First of all, did the whole Laboratory the last author of the manuscript, moreover in a role of corresponding author ? please complete the author list’
Answer: All authors participating in this experiment have been listed. There is only one corresponding author. Also, should we list all the authors of the lab (including those who did not participate in the experiment)?
Q2: Please explain the phrases, which were mentioned in the text first time: KWT3 (calmodulin is not enough), MS/MS analysis.
Answer: We accept the reviewer’s suggestion. The protein number identified by the MS/MS analysis was A0A0A0KWT3, which belonged to calmodulin-7 of Cucumis sativus (cucumber) based on the NCBI and the cucumber genome database.
Q3: In results – between line 7-11 failed the citation of the figure or supplement.
Answer: Per the reviewer’s comment, we have re-added Figure S1 and additional details to the manuscript.
Q4: Results, line 17 – authors declare citation figure 1 ,which is presented in the manuscript, but simultaneously add to them 3 citation - there is a problem, about which I have reservations. These type of presenting data’s suggest that figure was cited from previous paper , for example, and it is not an original result, so please adapt the figure;
Answer: The experimental method for the transient agroinfiltration of cucumber cotyledons was established in a previous experiment. However, the overexpression vector was pRI101-GFP in this manuscript, whereas the overexpression vector in the previous manuscript was pCAMBIA 3301 vector with luciferase (LUC). The method of experimentation is the same but the use of the vector is different. Therefore, Figure 1 is also different from that of the previous manuscript and represents an original result.
Q5: Figure 1 – confocal microscopy of the GFP-constructs presented in panel C –in panel 35S-GFP and 35S=CsMLO1-GFP authors have signal in cytoplasm and plasmalema, they did not notice the nucleus signal. Here, authors did not have a evidence [in this panel] of membranes, the resolution is definitely too small for that kind of conclusion.
Answer: Because they are not at a cellular level, the cell membrane, cytoplasm, and nucleus cannot be displayed together in the protoplast. In addition, we have added Figure S2 showing the localized tobacco leaf in the manuscript. The results demonstrated that CsMLO1 and CsMLO2 were localized in the plasma membrane.
Q6: Please explain in the materials and methods the method of measure the disease index, because there is no information about it
Answer: Per the reviewer’s comment, we have revised these omissions in the manuscript. Detailed experimental steps have been added to the materials and methods section (4.1).
Q7: The authors have a real problem with distinction Cucumber cotyledons from leaves – for example in Figure 11 (A) there are for sure the cotyledons ! not leaves! Authors stated that they described and comprised identification of CsMLO1-silencing effect on expression different defense-related genes 5days after treatment – The question is did the authors make the expression in cotyledons or at last in leaves, and the second question is did the authors realized that these expression in cotyledons and leaves could be completely different?
Answer: We accept the reviewer’s suggestion and have modified the relevant error description in the manuscript. All transgenic experiments were carried out in cucumber cotyledons in this manuscript. About the second question, we agree with the reviewer's opinion, although experiments on the stable transformation of cucumber are currently very difficult. Therefore, some experiments can only be performed in cucumber cotyledons.
Q8: Figure 3 and 11 it should be comparable scale of relative expression, then the focus on the differences will be more readable, at the moment it looks like rescale charts, unfortunately;
Answer: Thank you for this suggestion. The three replicates (three replicated columns) in Figure 3 showed a large difference; therefore, we re-performed this set of experiments. Data with similar relative expression are selected for processing, and the figure has been reconstructed. In addition, we also rescaled Figure 11. The relative expression of the three replicates of each of the CsPO1, CsRbohD, CsRbohF, CsPR1, CsPR3 and CsCaM3 genes was increased in the TRV::CsMLO1 cucumber cotyledons. The results indicated that CsMLO1 silencing enhances cucumber resistance to C. cassiicola infection.
Q9: Figure 4 the LUC image is completely unreadable, it is important and interesting analysis, therefore the micrograph should be enlarged; the same situation with figure 4 C;
Answer: We thank the reviewer for their comments. We have rebuilt the figure and inserted it into the manuscript.
Q10: Analysis from Figure 9 – what should be the reason of a very high differences between different samples plants in the same “treatment-condition”-especially in CsCaM3 in PO1, PR1 and PR3 ?? Moreover, please ensure that the asterisks of significance are visible;
Answer: We thank the reviewers for their comments. We think the reason for the considerable differences between different sample plants under the same "treatment conditions" may be due to the fact that the multiples of gene expression are different in different sample plants. Overall, the expression of PO1, PR1 and PR3 was up-regulated in CsCaM3-overexpressing cucumber compared to the controls. Notably, the extent of the increase in the transcription levels of these genes was significantly inhibited after coexpression of CsMLO1 and CsCaM3. Collectively, these results suggested that CsMLO1 overexpression suppressed CsCaM3-regulated defense gene expression and ROS burst. In addition, we have rebuilt Figure 9 and inserted it into the manuscript.
Q11: The general question is, Why authors choose exactly these ROS related genes to analysis like PO1, RbohD, RbohF ? Reviewer believe that these fungus induced ROS and PR, but why exactly these genes are important in immunity to Corynespora? there is no information about it, before the mentioned in discussion line 14
Answer: We thank the reviewers for their comments. In our previous study, transcriptome and iTRAQ analyses showed that the identified genes/proteins are mainly involved in defense response, oxidative stress and calcium signaling pathways in cucumber during C. cassiicola infection. In addition, our research found that the functions of CsMLO1 and CsMLO2 in C. cassiicola infection act as negative modulators to enhance the expression of ROS-related genes and defense-related genes for improved cucumber disease resistance. Thus, we explain that these genes, including ROS-related genes and defense marker genes, are important in cucumber immunity to C. cassiicola infection in the introduction.
Q12: Figure 7- once more there are cotyledons not cucumber leaves ! Moreover, authors used A nad B in the figure legend, but there is not information about it in the figure (??)
Answer: Your suggestion is correct. We have modified the relevant error description in the manuscript. In addition, A and B in the revised Figure 7 legend have also been modified.
Q13: Analyses in Figure 8 – authors used statements – evidently increased ,or evidently decreased, but the visualization in UV and light microscopy did not provide these kind of information, authors did not provides quantitative analysis of localization. Photographs in figure 8 are too expanded to conduct observation. Furthermore, there were visible no differences in three localization between transgens (A) and infected C. cassicola (B) – for example between CsMLO1 mock & infected or/and CsMLO1/CsCaM3 mock & infected. So, data about decreasing and increasing based on this figure are too speculative;
Answer: We thank the reviewers for their comments. We wanted to explain that cell death, callose deposition and accumulated levels of H2O2 was significantly increased in CsCaM3-overexpressing cucumber cotyledons, although the coexpression of CsMLO1 and CsCaM3 suppressed cell death in the transgenic tissues (A). Similarly, cell death, callose deposition and accumulated H2O2 levels were significantly increased in CsCaM3-overexpressing cucumber cotyledons, whereas the coexpression of CsMLO1 and CsCaM3 suppressed cell death in these tissues under C. cassicola infection (B). We did find such an experimental phenomenon in our experimental data.
Q14: Authors cited your own paper [citation 15] , but they not provided neither vol. nor pages or even the year of the publication;
Answer: We apologize for this mistake. We cited our own paper that has not yet been published. However, the relevant experimental results have to be cited here again. We have labeled this citation as "unpublished" [citation 15]. Please let us know whether this is appropriate. In addition, the reference [citation 13] has been accepted and will be published in the near future.
Q15: Composition of the manuscript, there is a problem, about which it has reservations. We have a lot of presented data –results page 2 to 15 and only 1.5 page of the discussion, which is shorter than introduction.
Answer: We thank the reviewers for their comments. We have partially supplemented the discussion section, which is not shorter than the introduction.
Reviewer 2 Report
In this manuscript, the authors combined a number of methodological approaches to show interaction of MLO1 and CaM3, which may have further consequences for the plant defense response. It is well written and I think the interpretation of results is supported by data. Some suggestions. The MIQE guidelines (Minimum Information for publication of Quantitative real-time PCR Experiments; Bustin et al 2009) proposes the standard abbreviation ‘RT-qPCR’ (not qRT-PCR) for reverse-transcription quantitative real-time PCR. Please, correct this over the manuscript. Figure 1B and others … When authors performed independent gene expression experiments, they show results for each of those experiments (with the SD as error measurement for each experiment). I think it would be more appropriate to show the average of the biological experiments with the corresponding standard error of the mean.
Author Response
Firstly, the issue of the article English language has been revisited through the Professional English Language Editing Company.
Q1: The MIQE guidelines (Minimum Information for publication of Quantitative real-time PCR Experiments; Bustin et al 2009) proposes the standard abbreviation ‘RT-qPCR’ (not qRT-PCR) for reverse-transcription quantitative real-time PCR. Please, correct this over the manuscript.
Answer: We accept the reviewer’s suggestion. The errors have been corrected in the revised manuscript.
Q2: Figure 1B and others … When authors performed independent gene expression experiments, they show results for each of those experiments (with the SD as error measurement for each experiment). I think it would be more appropriate to show the average of the biological experiments with the corresponding standard error of the mean.
Answer: We thank the reviewer for this suggestion. We have corrected these errors in the revised manuscript.
Round 2
Reviewer 1 Report
The manuscript has been significantly improved and now can be accept for publication in IJMS